# Stakeholder Perspectives on the Acceptability, Design, and Integration of Produce Prescriptions for People with Type 2 Diabetes in Australia: A Formative Study

**DOI:** 10.3390/ijerph21101330

**Published:** 2024-10-08

**Authors:** Kristy K. Law, Kathy Trieu, Jennifer Madz, Daisy H. Coyle, Kimberly Glover, Maoyi Tian, Yuze Xin, David Simmons, Jencia Wong, Jason H. Y. Wu

**Affiliations:** 1The George Institute for Global Health, University of New South Wales, Sydney, NSW 2000, Australia; ktrieu@georgeinstitute.org.au (K.T.); dcoyle@georgeinstitute.org.au (D.H.C.); kglover@georgeinstitute.org.au (K.G.); maoyi.tian@hrbmu.edu.cn (M.T.); jwu1@georgeinstitute.org.au (J.H.Y.W.); 2Diabetes Australia, Sydney, NSW 2037, Australia; 3School of Public Health, Harbin Medical University, Harbin 150081, China; yuze_xin@163.com; 4School of Medicine, Western Sydney University, Campbelltown, NSW 2560, Australia; da.simmons@westernsydney.edu.au; 5Macarthur Diabetes Endocrinology and Metabolism Service, Camden and Campbelltown Hospitals, Campbelltown, NSW 2560, Australia; 6Diabetes Centre, Royal Prince Alfred Hospital, Camperdown, NSW 2050, Australia; jencia.wong@health.nsw.gov.au; 7Faculty of Medicine and Health, Sydney Medical School, Central Clinical School, Central Sydney (Patyegarang) Precinct, University of Sydney, Sydney, NSW 2006, Australia; 8School of Population Health, University of New South Wales, Sydney, NSW 2052, Australia

**Keywords:** qualitative research, nutrition, food insecurity, healthcare, Food is Medicine, type 2 diabetes, social prescription

## Abstract

Produce prescription programs can benefit both individuals and health systems; however, best practices for integrating such programs into the Australian health system are yet unknown. This study explored stakeholders’ perspectives on the acceptability, potential design and integration of produce prescription programs for adults with type 2 diabetes in Australia. Purposive sampling was used to recruit 22 participants for an online workshop, representing six stakeholder groups (government, healthcare service, clinician, food retailer, consumer, non-government organisation). Participant responses were gathered through workshop discussions and a virtual collaboration tool (Mural). The workshop was video-recorded and transcribed verbatim, and thematic analysis was conducted using a deductive–inductive approach. Stakeholders recognised produce prescription as an acceptable intervention; however, they identified challenges to implementation related to contextuality, accessibility, and sustainability. Stakeholders were vocal about the approach (e.g., community-led) and infrastructure (e.g., screening tools) needed to support program design and implementation but expressed diverse views about potential funding models, indicating a need for further investigation. Aligning evaluation outcomes with existing measures in local, State and Federal initiatives was recommended, and entry points for integration were identified within and outside of the Australian health sector. Our findings provide clear considerations for future produce prescription interventions for people with type 2 diabetes.

## 1. Introduction

Type 2 diabetes (T2D) is a major public health issue in Australia. In 2023, T2D accounted for 2.2% of the total burden of disease, placing a significant burden on the healthcare system, economy and individuals [1,2]. The prevalence of T2D in Australia has continued to increase over the last decade and disproportionately impacts those living in areas of greater socioeconomic disadvantage [3]. Lifestyle modification, such as improving diet quality, is a critical step to support the management of T2D and reduce the risk of complications such as cardiovascular disease [4]. While consuming a healthy diet is a cornerstone recommendation for individuals with T2D, it remains difficult to achieve for most [5], particularly those experiencing food insecurity [6]. Food insecurity negatively impacts diet quality and mental health and has been associated with an increased risk of cardiometabolic diseases [7]. In 2020–2022, an estimated 11.4% of Australian households were food insecure [8]. Substantially higher prevalence has been reported among specific subpopulations facing socioeconomic disadvantage, such as single adult households [9], asylum seekers [10], Aboriginal and Torres Strait Islanders [11,12] and university students [13]. Despite the rising prevalence of T2D causing tremendous health, social and economic costs [14,15,16], there are limited evidence-based and scalable healthcare programs that can improve the diets of those with T2D.

“Food is Medicine” (FIM) is a social prescription approach to patient care that has gained rapid traction in recent years, predominantly in the United States (US) [17,18]. The approach encompasses a range of food-based interventions integrated into the funding and structure of healthcare systems for the prevention and management of diet-sensitive diseases, particularly among lower-income or food-insecure populations. Healthy food prescription (also called ‘produce prescription’) is one such strategy, which allows health professionals to ‘prescribe’ healthy food for vulnerable patients, often in combination with nutrition education and related social support to synergistically improve health behaviours [19]. Initial evidence about the effects of produce prescription is promising, suggesting reduced food insecurity and improvements in diet behaviour and metabolic risk factors [19,20,21,22]. When connected with local food producers and retailers, produce prescription has the additional benefit of supporting the local economy [23]. A prior pilot study in Australia demonstrated produce prescription’s considerable potential benefit for those with T2D, associated with substantial improvements in diet quality, blood lipid levels, and reduced body weight [24].

Several randomised controlled trials (RCT) are underway to further examine the efficacy and cost-effectiveness of FIM interventions for T2D and other diet-sensitive diseases [25,26,27,28]. While these RCTs are needed, best-practice models and frameworks for their implementation also require investigation. Complex interventions, such as FIM programs, are delivered across multiple levels (e.g., individual and organisational); therefore, multi-stakeholder engagement is a key element in their development and evaluation [29]. To ensure optimum translation of a complex intervention from research into practice, researchers must go beyond the question of effectiveness and gather diverse stakeholder perspectives on implementation factors, such as acceptability and scalability [29,30]. This is recommended practice not only for the development of new interventions but also for the adaptation of existing interventions to a new context or population [29,31]. Simply replicating an existing intervention in a new context may be ineffective due to differences in target population characteristics, implementation partners and/or local circumstances [31].

It would be remiss not to acknowledge the difference between the US and Australian health and social systems, policy and practice contexts. While new RCTs are underway, most FIM research studies to date have been small-scale pilots or quasi-experimental studies in the US. However, this has not dampened the acceleration of political and financial commitment to FIM programs across the US, enabling their rapid implementation in recent years. In contrast, FIM is an emerging concept in Australia and is yet to be adopted into healthcare practice. There is still a need to (1) build support among Australian stakeholders for the FIM approach as a strategy to address the negative impacts of food insecurity on health, (2) build the case for evidence-based policy to fund FIM interventions such as produce prescription in Australia, and (3) generate best-practice implementation models fit for the Australian health system context.

There is a gap in understanding how produce prescriptions would fit into the Australian health system, given the differences between US and Australian policy, healthcare funding models, and population characteristics. Most prior research examining stakeholder perspectives of produce prescription implementation has been conducted in the US [32,33,34,35,36], with very little data available in other countries, including Australia. In addition, these prior studies reported stakeholder perspectives for the purpose of evaluating an existing intervention [32,33,34,35,36] rather than exploring stakeholder perspectives to inform the development and implementation of a prospective intervention [29]. Research is needed to explore the perspectives of key stakeholders who would be involved in the implementation of produce prescription in Australia to identify what components may need to be adapted for the local context [29,31].

Therefore, this study aimed to generate context-specific evidence to support the integration of produce prescriptions in Australia. Specifically, we explored Australian stakeholders’ acceptance of and perspectives on the design and implementation of produce prescription interventions for people with T2D. We sought to address the following research questions:What is the acceptability of produce prescriptions for T2D to key stakeholders in Australia?What might a produce prescription program look like for individuals with T2D in Australia?How would key stakeholders integrate produce prescription as a ‘therapy’ for T2D, considering existing T2D prevention pathways and treatment models?

## 2. Materials and Methods

This qualitative study represents the first stage of a multi-stage partnership research project which aims to co-design a sustainable model for implementation partners and a framework for produce prescriptions in Australia. The larger research project will evaluate the clinical impact of a produce prescription intervention on individuals with T2D experiencing food insecurity through an RCT [37]. Ethics approval for this study was granted by the University of New South Wales Human Research Ethics Committee (HC No: HC230258) and is reported here following the Consolidated Criteria for Reporting Qualitative Research (COREQ) guidelines [38].

### 2.1. Research Team and Reflexivity Statement

This research study was part of a partnership initiative between Diabetes Australia, a national peak body committed to reducing the impact of diabetes, and The George Institute for Global Health (The George Institute), a not-for-profit global medical research institute at the University of New South Wales Australia focused on non-communicable diseases prevention and treatment. Together, the study team consisted of individuals with diverse research, policy, and/or practice experience in nutrition, food policy, health systems, clinical trials, and implementation science. Recognising these research interests, regular meetings were held to allow discussion of individual team members’ perspectives on the study design, with the assistance of an independent facilitator. Throughout the analysis process, the researchers practised reflexivity by reflecting on emerging themes and their own values and perspectives, acknowledging any potential influence on data interpretation and concluding findings.

### 2.2. Study Design

The overall approach for this study was informed by the literature on collaborative research approaches and the development of complex research interventions [29,39,40]. The multi-stakeholder workshop method was considered appropriate for this study, given our formative research focus [41,42]. Prior research has also suggested that the workshop method offers important strengths, including helping to promote consultative and collaborative participation, facilitating in-depth deliberation and shared understanding of an issue, and is useful when developing or evaluating new services [41].

### 2.3. Participant Selection

The informant categories outlined in Auvinen, Simock, and Moran’s study were adapted to the Australian context to identify the target stakeholder groups who would be potentially involved in the implementation of produce prescriptions in Australia (Table 1). Key individuals from these target stakeholder groups were then identified through purposive and snowball sampling based on their expertise and experience, including the following individuals:Those in positions of implementing and planning healthcare programs in Australia;Those managing food retail services relevant to producing prescription program models;Those with lived experience of T2D and/or having experienced food insecurity;Those who are health advocates focused on the health and well-being of those with T2D.

Stakeholders were identified and approached by the research team via existing networks from different sectors, including local government, state and federal government policymakers, the food industry sector, non-government organisations, and academia. Direct recruitment methods were used to ensure a balanced presence of individuals across stakeholder categories and to select stakeholders with the ability to provide expert insight relevant to the research questions under investigation. Targeted stakeholders could also nominate and recommend additional clinicians and/or consumer representatives to the research team who they thought could provide insight relevant to the research project. All targeted and recommended stakeholders (*n* = 25) were sent a letter of invitation via email to become members of the project’s Translation Advisory Group (TAG). As part of accepting the invitation, stakeholders agreed to participate in workshops and other co-design process activities over the course of the broader multi-year research project, with the first workshop forming the basis of the current research paper.

### 2.4. Data Collection

The TAG workshop was hosted online by The George Institute on 8 August 2023. Prior to the workshop, all participants completed informed consent. All consumer representatives were remunerated for their participation [43]. The 90-minute virtual workshop was conducted using Microsoft Teams and facilitated by a member of the project team (JM) who had extensive experience in stakeholder engagement across both public and private sectors and was not involved in data analysis.

To support stakeholders’ input into the workshop, a background brief was developed via a rapid review and preliminary synthesis of the literature to present produce prescription case studies, which was provided as pre-reading material. The workshop covered three topics overall (i.e., acceptability, program design and implementation; program integration) that were related to the study’s research objectives (Appendix A). To guide discussions, workshop questions were developed and adapted from the existing literature. Given the formative research focus, acceptability was a concept deemed suitable to assess, in line with Klaic et al.’s framework of implementability [30]. Questions around program design and integration were specifically informed by papers that examined produce prescription implementation practices and stakeholder perspectives [35,36,44,45]. All questions were pilot-tested prior to the workshop.

Participant responses during the workshop were gathered using a web platform called Mural. Mural is a virtual whiteboarding tool that enables collaboration by allowing for visual brainstorming of solutions/problems and challenges (https://www.mural.co, accessed on 8 August 2023) [46]. A demonstration of how to use Mural was provided by the facilitator to all participants at the start of the workshop, followed by an opportunity for participants to practice using the tool to create a response. A member of the research team was also appointed as a moderator to address any technical difficulties that arose during the workshop. Participants provided their responses to all workshop questions in writing by creating virtual ‘sticky notes’ in Mural. In addition, conversation prompts were offered by the facilitator to encourage group discussion and verbal responses to topics raised during the workshop. Participants could also answer questions and contribute to the discussion by using the Microsoft Teams chat function, which provided an alternative option for participants who experienced technical difficulties using Mural. Only one participant reported having difficulties using Mural and instead provided input via the Microsoft Teams chat function.

Stakeholders who accepted the invitation to participate in this study but were unavailable to attend the workshop were sent a survey via Qualtrics featuring the same set of questions to enable them to provide written feedback on the workshop topics (Appendix A). Most of the survey questions were free-text responses; other question types included single answers, multiple choice, and Likert scale items. The survey was pilot-tested prior to distribution.

### 2.5. Data Analysis

Three authors conducted the data analysis for this study (KKL, KT, JHYW). The workshop was video recorded with permission from participants and transcribed using Microsoft Teams 365. The transcription file and the Microsoft Teams chat log were verified against the workshop recording for data authenticity and consistency by one researcher (KKL). Data collected via Mural and the online survey were exported into Microsoft Excel 365 and checked for typing errors by the lead author and a research assistant.

All data were de-identified and imported into NVivo 12 (Lumivero, Denver, CO, USA) for thematic analysis using a deductive–inductive approach. A codebook was developed in Microsoft Excel by one (KKL) using the three workshop topics as a priori domains, with discussion questions guiding the generation of initial codes. Code summaries and preliminary workshop findings were collated into a summary report for TAG members, which was circulated to provide an opportunity for further comment or clarification. Inductive analysis was then conducted by one researcher (KKL), using exploratory coding to build upon the codebook and generate initial themes across the dataset. Initial themes were reviewed with two additional co-investigators (KT, JHYW) to ensure cohesion, with differences resolved by discussion. These themes were then further refined via discussions among three researchers (KKL, KT, JHYW) to reach a consensus on the final themes.

## 3. Results

### 3.1. Stakeholder Characteristics

Twenty-two stakeholders accepted the invitation to participate in this study, representing government (*n* = 5), healthcare services (*n* = 5), clinicians (*n* = 4), food industry (*n* = 4), consumers (*n* = 2), and non-government organisations (*n* = 2) (Table 1). Stakeholders were based in New South Wales (NSW) (*n* = 16), Western Australia (WA) (*n* = 4), and the Australian Capital Territory (*n* = 2). On the day of the workshop, one stakeholder provided feedback via the Qualtrics survey due to unexpected illness, whereas the remaining 21 stakeholders attended the virtual workshop as planned.

Three stakeholders declined the invitation to participate in this study due to scheduling conflicts (*n* = 2) and illness (*n* = 1). These stakeholders were all based in NSW.

### 3.2. Synthesis of Findings

Nine themes were identified from stakeholders’ perspectives across this study’s three research questions (Table 2) and are described below.

#### 3.2.1. Research Question 1: Stakeholder Acceptability of Produce Prescription for People with T2D

**Produce prescriptions can address upstream determinants of health that improve a broad range of health outcomes.** All stakeholders agreed that the concept of produce prescriptions aligned with their personal values and/or organisation goals to improve and maintain the health of their communities served. Overall acceptance was intrinsically linked with the perceived potential of produce prescriptions to provide multiple health benefits beyond just diabetes-specific outcomes. Potential benefits identified related to the individual recipients (e.g., improved diet quality, improved chronic disease self-management) and the healthcare sector (e.g., meets a gap in clinical care, reduced healthcare spending). Other benefits identified related to the creation of supportive systems (e.g., via influence on family eating patterns), promotion of equitable access to healthy food, and supporting Australian fruit and vegetable growers.

**Produce prescription implementation faces contextual, accessibility, and sustainability challenges.** While the concept of produce prescriptions was acceptable to all stakeholders, potential challenges to their broader acceptability and implementation in Australia were also identified. Stakeholders’ responses fell into three main categories: contextuality, accessibility and sustainability challenges.

Contextuality: To aid successful engagement and implementation, stakeholders expressed the need to ensure produce prescription programs were fit for purpose. Stakeholders perceived that a variety of factors related to the recipients (e.g., cultural needs, household composition) and/or implementation context (e.g., logistical infrastructure, rurality) could impact produce prescription acceptability. Therefore, programs need to be designed to fit the needs of individuals and their broader contexts. One participant summarised as follows:


*“…understanding why we’re doing something and understanding what the guiding principles are (is) really important because the same rules and the same solutions you know, be it a home delivery service or a community garden that work in one part of even this local health district, won’t work in another.”*
(Healthcare Services 1)

Accessibility: Stakeholders mentioned the need to consider the ease at which participants could receive and use the produce provided in their local context. Common barriers to accessibility and uptake were identified, such as stigma, risk of theft, food safety (i.e., both in terms of supply and at-home storage), serviceability (particularly in rural and remote areas of Australia), personal factors (e.g., cooking skills, time, motivation), and medical contexts, such as the following, for example:


*“Patients’ disability; many patients’ with existing metabolic conditions may also concurrently have disabilities, especially in movement, as well as mental health such as depression, that affect their daily mood and behaviour. All of this could impact their uptake of daily produce prescription.”*
(Healthcare Services 5)

Sustainability: The sustainability of produce prescription was a key challenge voiced by stakeholders, with queries raised around the availability of funding and resources to support sustainable program implementation after the RCT is finished. Stakeholder responses highlighted the need for integration into funders (e.g., State government or universal health insurance schemes such as Medicare) and main public healthcare providers (e.g., Local Health Districts or Primary Health Networks) to ensure the sustainability of the implementation of this program.

Conversely, other respondents acknowledged that obtaining funding is likely to be challenging and complex, particularly within the government context and highlighted the need for alternative funding streams, for example:


*“Initial funding to kickstart the program should come (from) State Government—but eventually this should be commercially sustainable and pay for itself. We need scale, and so far, NSW Government has not been forthcoming with funding for prevention. They are more interested in funding medication. Also, if you give things away for free, people tend to not value it beyond the first few weeks or months.”*
(Food Retailer 1)

These differences in perspectives may reflect the differences in participants’ experiences with public funding and program implementation between sectors. Other challenges to produce prescription sustainability raised by stakeholders included the longer-term sustainability of behavioural improvements made by participants during the program (e.g., dietary changes, adherence to medical advice); and the collection of data (e.g., cost-effectiveness outcomes) needed to build the case for produce prescription integration into health systems and other existing services.

#### 3.2.2. Research Question 2: Produce Prescription Program Design for People with T2D

Four themes were identified regarding the design and implementation of a prescription program for people with T2D. Stakeholders were asked to consider the following program components: the prescription/incentive; screening and referral pathways, delivery mechanisms, implementation, monitoring, and evaluation.

**Produce prescription requires multisectoral partnerships governed by clear structures.** Partnerships across various sectors and levels would be required to design and implement a produce prescription program. Potential program partners were identified from six main sectors, with many stakeholders suggesting the following specific examples: academia; non-government organisations (e.g., Rural Doctors Network); consumers and community-based organisations (e.g., Carers WA, churches); government; healthcare; and the corporate sector (e.g., food transport companies). Stakeholders identified six key roles overall that program partners could fill: coordination and logistics; screening and referral; funding; advocacy and promotion; food provision; and education. Gaining consensus on partner-role allocations was beyond the scope of the workshop. The role of program partners was an important topic to some stakeholders, as evidenced by their questions posed in need of further discussion. For example, one stakeholder said the following:


*“Who implements it? At what level? Who “owns” it? How do you manage it on such a broad scale?*
(Healthcare Services 1)

Most stakeholders expressed the need to establish formal governance structures for clear accountability. This was most frequently suggested to be a representative governance body comprised of key partner agencies and consumers. However, stakeholders also mentioned other structures that would be needed to support this governance body, such as partnership agreements, service delivery contracts, and a program coordination group:


*“There will need to be a co-ordinating group or organisation. The group could be made up of representatives from the agencies, companies etc involved. Or there might be a specific agency or non-government organisation established to run this—would need funding. There would need to be local coordinator or manager who worked under the guidance of the co-ordinating group or agency”*
(Healthcare Services 4)

**Produce prescriptions should be directed by shared goals and genuine collaboration between partners.** In discussions around ways of working, there was a clear distinction between *what* stakeholders thought partners should do and the approach for *how* partners should work together. To facilitate collaborative ways of working between partners, stakeholders expressed the importance of developing a shared vision. Some stakeholders provided suggestions on what the initial steps would entail, such as *“first identifying how to align strategic priorities”* (Non-Government Organisation 1); developing a *“joint statement of intent for organisations involved”* (Government 4) or using a *“co-commissioning approach”* (Government 3).

Some stakeholders also considered it important to define the nature of the partnership approach, particularly when working with target communities and during program implementation. While one stakeholder simply summarised it as *“Talk with patients, not at them”* (Consumer 2), others described the need for genuinely collaborative and place-based ways of working. One stakeholder shared the following:


*“…Rather than saying, look, this is what the policy says…this is what we’re going to implement across the State based on fixed rules, I think social solutions like this really need to be place based and so that’s where adhering to principles and values, rather than rules, is really important.”*
(Healthcare Services 1)

**Produce prescription design and implementation needs to be contextualised and community-led.** Stakeholders were asked a series of questions to explore their perspectives on how each program component would be designed to a produce prescription for adults living with T2D in Australia. This provoked a rich response from stakeholders. Context emerged as the most significant factor that needed to underpin every component of produce prescription design and implementation (Table 3). Stakeholders emphasised how a one-size-fits-all approach was unlikely to succeed, instead suggesting a suite of options tailored to the needs of the recipient, local community, and existing systems in place.

The importance of context was illustrated clearly by one stakeholder when considering the delivery of such programs in their own local health district’s location:


*“Most areas in Western New South Wales Local Health District (are) located in rural and remote areas of Australia with a clear supply chain problem of fresh produce and limited options of big chains supermarkets such as Coles, Woolworths and Harris Farm. This also creates a sustainability issue that will likely hamper the implementation of this type of program....”*
(Healthcare Services 5)

**Produce prescription should target priority populations with T2D through non-stigmatising approaches.** Most stakeholders agreed that a T2D produce prescription should prioritise the screening and referral of people experiencing greater socioeconomic disadvantage, pregnant women, and/or those with comorbidities (e.g., depression, obesity). In contrast, other stakeholders highlighted the importance of self-determination and felt that there should be a pathway for *“Anyone who would like to participate for health gain whilst living with diabetes”* (Government 1). Another stakeholder said the following:


*“There ideally would be an option for a non-referred service where people can put their hand up, identify this is their need, and not have the barrier of needing to get a referral from someone.”*
(Healthcare Services 1)

Suggestions for a non-referred, open service is an interesting finding, considering the current levels of health system spending and resource constraints in Australia. This may reflect the values held by some stakeholders around the approach to patient care. Stakeholders were vocal about the need for simple, patient-centred screening and referral processes and highlighted the importance of a trauma-informed, non-stigmatising approach:


*“Screening should be as simple and non-stigmatising as possible. Consider the environment and way in which the question will be asked. It is a sensitive topic. Referral should be done then and there by the health professional. The more complicated or convoluted the process, the less likely the individual will engage.”*
(Clinician 4)

#### 3.2.3. Research Question 3: Pathways for the Integration and Adoption of Produce Prescription for T2D into the Australian Healthcare System

The final part of the workshop asked stakeholders to think about how they could see produce prescription as a ‘therapy’ being integrated into a model of care for T2D specific to their context of work (i.e., at a local/district/state level). Three themes were identified from stakeholder responses to this question.

**Entry points for produce prescription integration exist within and outside of the health sector but may need new tools or technologies.** Several entry points for the integration of produce prescription programs for T2D were suggested both within and outside of the healthcare sector. However, stakeholders noted that new tools and technologies would be needed to integrate produce prescriptions successfully. Within existing healthcare, stakeholders raised the following:


*“Clinical Dietitians in hospital can screen like they do with malnutrition screening and then refer on as part of discharge papers.”*
(Government 5)


*“Should be free for participants linked to ongoing health condition management with a care coordinator, i.e., as long as care is being provided, this forms part of the care.”*
(Government 1)

However, additional actions would be needed to support healthcare professionals to act as program referrers, such as the following:Develop referrer training, education, and practice guidelines;Provide more clarity about program criteria and evidence of benefits;Develop brief referral processes and clear action pathways;Build new electronic resources (e.g., assessment tools, Medicare item number) into existing software or medical records;Build linkages into community networks and services.

Several entry points outside the healthcare system were also identified, such as through schemes offered by other government departments (e.g., Veterans Affairs, Aged Care), or services provided by the community (e.g., cultural groups) or charitable food sectors. Such entry points could improve the reach and engagement of produce prescription programs, as one stakeholder explained:


*“Expanding the range of referrer roles will increase the chances of capturing patients who don’t have a strong connection with the health system already and could possibly have the greatest benefit.”*
(Government 3)

**Potential funding depends on program design factors and participant eligibility criteria; therefore, a mixture of models should be considered.** No single funding model or arrangement dominated the discussion, with a diversity of opinions expressed (Table 4). Again emphasising the influence of contextual factors, stakeholders suggested that the appropriate funding model would heavily depend on how the program was designed and rolled out. Stakeholders who supported publicly funded models felt that other funding models (e.g., private health) would not meet the needs of people with food insecurity and viewed the government as having *“a responsibility to care for the health of its citizens”* (Clinician 4).

**Align outcome measures with high-level strategies and local priorities.** Stakeholders identified a range of outcomes that should be evaluated for a T2D produce prescription program, which fell into four main categories: health outcomes (e.g., wound healing, patient-reported outcomes), health economics, program evaluation (including food quality), social (e.g., financial security), and long-term impact measures. The need to align proposed program measures with existing outcomes in other initiatives at the local, State, and Federal levels was emphasised by several stakeholders. For example, aligning produce prescription outcomes with the NSW Statewide Initiative for Diabetes Management would support the NSW Ministry of Health’s commitment to value-based healthcare [47] and help *“prepare for future comparison or scale of programs”* (Government 3).

While the workshop question focused on the ‘what’ of monitoring and evaluation, some stakeholder responses provided insights into the proposed approach to monitoring and evaluation. Perspectives included that monitoring efforts should aim to foster shared learning across different localities and settings:


*“I would love to see a level of localised community monitoring that is then able to be compared, in an information sharing and collaboration rather than benchmarking nature, to help identify areas that could learn from other communities, communities who could do with extra support, communities doing novel things or where there needs to be greater government investment.”*
(Healthcare Services 1)

## 4. Discussion

This qualitative study is the first to investigate Australian stakeholder perspectives on produce prescriptions for T2D to guide future design and implementation. Acceptability is recommended as one of the first implementation concepts to assess when developing an intervention [30]. While there was strong acceptance for produce prescription expressed across a range of stakeholders (e.g., government, healthcare services, food retailers), concerns were raised about the perceived contextual and accessibility challenges to implementation. Our findings confirm the need to design tailored, produce prescription programs that address these barriers; however, questions remain around how to do this while achieving feasibility and scalability. Further research to explore these issues would also help address concerns raised by stakeholders around the likelihood of trial participants sustaining any changes made to their food consumption habits once the produce prescription RCT ends.

Another main concern was the need for funding to support the long-term sustainability of produce prescription programs. However, stakeholders held differing views on which potential funding models (e.g., public or private) would best achieve this. Our findings highlight the need for further investigation into funding considerations, including participants’ willingness to pay [48] if produce prescriptions are to be sustainably embedded within routine care for T2D. Stakeholders identified several opportunities for produce prescription integration in Australia, both within (e.g., specialist metabolic health clinics) and outside of the healthcare sector (e.g., via partnerships with food banks). Developing referral infrastructure and aligning program measures with outcomes in existing initiatives were perceived by stakeholders as key factors for successful integration.

### 4.1. Potential Implications for Practice

The insights gained from this study offer several implications and potential uses for practice. Firstly, study findings will be used to inform an integrated and coordinated implementation model for the larger research project’s produce prescription RCT for people with T2D. For example, a key characteristic of FIM strategies like produce prescriptions is supporting recipients’ access to food through a healthcare provider referral [36]. While Australian stakeholders agreed this would work as a primary mechanism, alternate referral pathways such as self-referral were also suggested. Given constraints on healthcare providers’ time and resources, the option of self-referral to produce prescription programs was a novel suggestion. We have integrated a self-referral pathway into the project’s RCT design via social media advertising, providing an opportunity to examine its feasibility in comparison to the commonly used clinical referral pathway.

Secondly, given the emphasis on context as a key consideration for produce prescription integration, future studies could apply implementation science frameworks in their design to gain a more nuanced understanding of predictors of successful implementation across diverse Australian contexts. Based on our findings, such an approach will be particularly important for the urban vs. rural/regional contexts in Australia due to its geography and varied availability of nutritious produce. Examples of potentially useful frameworks from prior research include the Exploration, Preparation, Implementation, and Sustainment Framework [49] used by Houghtaling and colleagues [50] to develop a pragmatic implementation checklist for FIM programs in healthcare settings and the Consolidated Framework for Implementation Research [51], which has been used to evaluate produce prescription programs [36,52] or guide their future development [27,53].

### 4.2. Putting Our Findings in Context

Only one other study has explored the perspectives of stakeholders in anticipation of a produce prescription program. Thomson et al. [54] explored the perspectives of Black and Hispanic individuals living in the US at risk of food insecurity on a potential produce prescription program. By eliciting community voice, this study provides detailed insights into the potential impact of produce prescriptions on the lives of those such programs are intended to reach. Similarly, their study emphasised the need to avoid a “one size fits all” approach and highlighted the need for community involvement before implementation to ensure that programs are tailored to each community’s unique context. Our study extends this prior research by being conducted in Australia and including additional stakeholders from broader stakeholder groups. While our research questions focused on T2D and produce prescriptions, the considerations raised by stakeholders in this study may also be useful for researchers or healthcare providers interested in developing FIM programs for other diet-sensitive conditions in Australia.

Our findings represent an initial roadmap to support such efforts by positioning FIM interventions as a way to achieve key objectives of existing initiatives, services, or schemes at local, State, and Federal levels. For example, a recent parliamentary inquiry report into diabetes in Australia recommended action be taken to “support access to healthy food to all Australian communities (Recommendation 6)” and improve “equitable access to healthcare for people living with all forms of diabetes (Recommendation 12)” [55]. Furthermore, there is alignment between FIM programs and the Australian Government’s Primary Health Care 10 Year Plan 2022–2032, which states that a key priority is to “improve access to appropriate care for people at risk of poorer health outcomes” [56] (p. 37) via agreed actions such as “to develop, refine and scale evidence-based models of social prescribing… for at-risk and disadvantaged groups.” (p. 37). This strategic approach to the continued investigation of produce prescription and other FIM initiatives in Australia may also support their sustainability.

### 4.3. Strengths and Limitations

Several strengths were built into our study. These include the following:Credibility—A summary report of initial workshop findings was shared with all stakeholders to provide them with an opportunity to clarify and/or verify the researchers’ interpretation of their responses. The report was accompanied by an invitation for further input, if stakeholders had additional information to contribute.Dependability—While the lead author generated the codes, all initial and final themes were reviewed by two co-investigators in this study.Flexibility—Involving stakeholders through co-production processes is recommended when developing complex research interventions such as produce prescriptions [29]. Nevertheless, traditional methods (e.g., in-person focus groups) can be costly or inaccessible for some stakeholders to attend, potentially impacting the diversity of insights gained [57,58]. Considering other commitments of the invited stakeholders, offering a flexible combination of virtual data collection methods (workshop and survey) provided a greater opportunity for participants to provide input and reduced respondent burden. These advantages have been commonly reported in prior studies using virtual qualitative methods and tools [57,59,60].Transferability—Additional representation from other jurisdictions across Australia would have been ideal to adequately explore the scalability of produce prescription programs. However, since the planned RCT will be performed in NSW, we deliberately over-sampled stakeholders from NSW, which enhances the likelihood that the findings are most generalisable to the NSW context and inform the design of our RCT.


Some limitations include our recruitment strategy, which relied on targeted invitations from the research team to stakeholders identified within their existing networks. While this approach assists with snowball sampling, it may have skewed our sample with stakeholders who were already in favour of produce prescription programs, introducing potential biases to our findings, such as to the question of acceptability. It is possible that our findings could have been influenced by the subjectivity of the opinion of the stakeholders we engaged with. Future studies in Australia engaging with different sets of stakeholders (e.g., those based in different states) could help to further validate our findings. The sample size may be considered a further limitation of this study, although small participant groups are a common feature of workshop methodology to promote balanced and genuine participation [42]. Furthermore, as an exploratory study, the sample size of 22 was deemed sufficient to address the research questions as it included individuals across a diverse range of stakeholder groups necessary for produce prescription implementation. Multiple representatives from local government, state policymakers, the federal government, the food industry sector, non-government organisations, and consumers participated in the workshop, adding to the richness of the data obtained. This recruitment strategy is also comparable to a prior research study examining the integration of produce prescription programs in the US, which included 19 key stakeholders [36]. Similarly, other multi-stakeholder health research studies have reported sample sizes ranging from 16 to 30 participants [61,62,63].

We also acknowledge this study’s methodological limitations. The workshop method is particularly useful in generating data to inform future processes, such as developing a new service [42]. However, only one 90-minute workshop was conducted for this study. In comparison to other qualitative research methods (i.e., semi-structured interviews), the workshop format and time restraints gave us less opportunity to probe for additional responses with each individual stakeholder. For example, the workshop left some questions in need of further investigation, particularly around governance and funding. These limitations were partly mitigated by opportunities for stakeholders to provide follow-up comments, as mentioned above. As part of the co-design process, future workshops could focus their scope on gaining stakeholder consensus on these important program components.

While conducting qualitative research in virtual environments has become increasingly popular, literature to specifically guide the conduct of online workshops and associated technology is still emerging [57]. As such, using a virtual tool (i.e., Mural) may have resulted in limitations not encountered with traditional methods. While whiteboarding tools such as Mural allow participants to simultaneously respond to the workshop questions, this potentially did not allow for in-depth discussion between participants, which may have been achieved through other methods (i.e., focus groups). However, we believe the potential benefits of using the interactive web platform justified its use. By allowing simultaneous responses, participants may have been more likely to share their own thoughts/opinions and less likely to be influenced by others, which may have mitigated any perceived power imbalances between stakeholders (e.g., senior government versus consumer) [42]. In addition, virtual tools have been found to facilitate the learning of new concepts via visualisation [57], help develop design thinking skills [64], and expedite data validation [58], which are all beneficial to the process of co-designing complex interventions.

Some participants had also never used Mural before the workshop. However, except for one participant, all others during the virtual workshop did not have any difficulty using the Mural tool. The one participant who was unable to use Mural provided extensive input via the Microsoft Teams chat function. This suggests that the strategies we implemented to mitigate this issue, such as providing a walk-through demonstration, on-demand technical support, and alternative methods for collecting verbal and written responses, were effective in curbing any technical difficulties that may have played a major role in causing bias in our results. However, we acknowledge that using a virtual tool may have limited a stakeholder’s ability to participate fully during the workshop.

## 5. Conclusions

The overall positive appetite for produce prescription among a range of stakeholders provides justification for continued investigation of such programs in the Australian context. Findings from this study will be used to inform the design of an RCT for people with T2D, and subsequent workshops have been proposed to continue to engage stakeholders to support the evaluation of the RCT findings. In addition, this study provides valuable insights into the design and implementation of produce prescriptions. A key consideration highlighted by stakeholders in this study was the importance and potential challenge of contextuality. Our study found that context considerations were relevant across all program components, from intervention design and delivery to funding models. These findings can be used to inform future research to support the RCT results and refinement of produce prescription programs.

Questions remain around the governance and sustainability of produce prescription, particularly after the RCT ends. While several potential pathways for integration were identified that could support sustainable produce prescription programs, future studies are needed to explore the specificities of how this could be achieved. Our findings represent an initial roadmap for produce prescriptions in Australia and demonstrate stakeholder support for the integration of FIM strategies into routine care for vulnerable individuals with T2D.

## Figures and Tables

**Table 1 ijerph-21-01330-t001:** Summary of produce prescription Translation Advisory Group stakeholders.

Stakeholder Group	Description of Expertise Related to Produce Prescriptions	Number of Stakeholders
Government ^1^	Familiarity with local, state, and/or federal healthcare and public health policy, procurement processes, program, and initiative funding mechanisms that may support produce prescriptions.	5
Healthcare Services	Administers and/or providers of healthcare and support services at the local level who receive funding to cover these services. Administration may include setting up parameters of health services/programs, coordinating payment and data collection among healthcare providers, managing system processes for program operations.	5
Clinician	Provides direct clinical care to their patients and understands clinical treatment workflow including referrals, guideline protocols, requirements of continuation of care, clinical data collection and reporting needs, funding models, and billing reimbursements—both public and private sectors.	4
Food Retailer	Oversees procurement, supply, and distribution of food to customers. Understands food system and vendor processes and systems that may be needed to operationalise produce prescriptions.	4
Consumer	Provides perspectives and experiences related to health care use, living or caring for those with diet-related health conditions, and health-related social risks (e.g., education, socioeconomic status).	2
Non-government Organization	Advocates for and contributes to policy development to improve quality of service delivery for their members. Provides insight into the political landscape at the federal and state levels for program implementation and scaling specific to its members.	2

^1^ Includes local, State and Federal government representatives.

**Table 2 ijerph-21-01330-t002:** Description of final nine themes by research question.

Research Question	Themes
1. What is the acceptability of produce prescription for T2D to key stakeholders in Australia?	Produce prescription can address upstream determinants of health that improve a broad range of health outcomes.
	Produce prescription implementation faces contextual, accessibility, and sustainability challenges.
2. What might a produce prescription program look like for individuals with T2D in Australia?	Produce prescription requires multisectoral partnerships governed by clear structures.
	Produce prescription should be directed by shared goals and clear, collaborative ways of working.
	Produce prescription design and implementation needs to be contextualized and community led.
	Produce prescription should target priority populations through non-stigmatizing approaches.
3. How would key stakeholders integrate produce prescription as a ‘therapy’ for T2D, considering existing T2D prevention and treatment models?	Entry points for produce prescription integration exist within and outside of the health sector but may need new tools or technologies.
	Potential funding sources depends on program design and participant eligibility criteria; therefore, a mixture of models should be considered.
	Align outcome measures with high-level strategies and local priorities.

**Table 3 ijerph-21-01330-t003:** How context underpins every component of produce prescription design and implementation with illustrative quotes from stakeholders.

Program Component	Illustrative Quotes
Screening and referral	“Create an assessment framework that is principles based and can be adapted to suit local context and cultural background.” (Government 3)
Program design	“We cannot have meaningful conversations with patients and expect them to engage if the approach is completely not fit for the context in which they live.” (Healthcare Services 1) “I think it is important to have a champion in the community to lead this program that might potentially come from community-based organizations.” (Healthcare Services 5)
Program delivery and implementation	“Ideally offer multiple methods of access e.g., home delivery and option to attend physical location (ideally multiple locations that are easily accessible).” (Clinician 4) “Using existing food distribution networks/retail environments…existing non-government services, need to consider patient ability to collect and store produce.” (Government 2)

**Table 4 ijerph-21-01330-t004:** Potential funding models for produce prescriptions in Australia.

Funding Model	Illustrative Quotes
Block funding	“If to be integrated into health system, would need to be block funding or part of funding for National Weighted Activity Unit.” (Healthcare Services 2)
Hybrid/Staged funding	“Needs a hybrid approach level 1—free, level 2—co pay, and co pay could rise.” (Government 4) “Consider a staged funding model—start with block, build to collaborative commissioning, then aim for commonwealth.” (Government 4)
Collaborative Commissioning—shared investment	“Seek out a collaborative model such as in Western NSW collaborative commissioning.” (Government 3)
Private health	“Health insurance companies may fund this as a prevention program.” (Government 5)
National Diabetes Services Scheme	“Through Primary Health Network space initially and ideally if a national approach, eventually National Diabetes Services Scheme and Medicare Benefits Schedule.” (Government 2)
Pharmaceutical Benefits Scheme or Medicare Benefits Schedule	“If implemented in primary care—Medicare Benefits Schedule or Pharmaceutical Benefits Scheme would be most logical.” (Healthcare Services 2)
Other	“A mixture of funding sources including government and philanthropic…including philanthropic organization as funders (whose values and goals align with produce prescription) could support its sustainability and expansion.” (Clinician 4) “Company Sponsorship or Donations” (Food Retailer 2)

## Data Availability

The data presented in this study are available on reasonable request from the corresponding author. The data are not publicly available due to ethical constraints.

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
