# Peer review of "Stakeholder Perspectives on the Acceptability, Design, and Integration of Produce Prescriptions for People with Type 2 Diabetes in Australia: A Formative Study"

_ijerph, 2024, doi:10.3390/ijerph21101330_

Round 1
Reviewer 1 Report
Comments and Suggestions for Authors
Introduction section:
1. While the introduction briefly mentions that most studies on produce prescriptions have been conducted in the United States, it does not sufficiently explain why exploring stakeholder perspectives specifically in the Australian context is crucial. Providing a more detailed rationale for focusing on stakeholder perspectives would strengthen the introduction and underscore the importance of the study.
2. The introduction could benefit from including more references to studies or data specific to Australia, particularly regarding food insecurity and T2D, to better ground the discussion in the local context.
3. The introduction mentions the limited data on produce prescriptions in countries outside the United States but could better articulate the specific gaps in the literature that this study aims to fill. A clearer outline of these gaps would provide a stronger foundation for the study's research questions.
Material and Methods section:
1. The research design is well-suited to the study’s objectives and is methodologically sound. It effectively incorporates stakeholder input, uses appropriate qualitative methods, and includes mechanisms to ensure the rigor and validity of the findings. The study’s design is capable of generating valuable insights that can inform the development of produce prescription programs in Australia. However, the study could further strengthen its design by addressing potential limitations, such as selection bias and the relatively small sample size, which could impact the generalizability of the findings. Additionally, a more detailed explanation of how the findings from this formative stage will be integrated into the larger research project could enhance the overall research design.
2. Although the sample size of 22 stakeholders is mentioned, a more explicit justification for why this number was deemed sufficient could be provided. This would help readers understand the rationale behind the sample size in relation to the study’s objectives.
3. it could benefit from a brief discussion of any challenges encountered during data collection or analysis, such as potential biases introduced by the sampling method or any technical difficulties with the virtual tools used.
4. A more explicit acknowledgment of the methodological limitations, such as the reliance on virtual tools for data collection and potential biases in participant selection, could enhance the critical appraisal of the study.
Results section
1. The results of this study are clearly presented, with a well-structured and logical organization that makes it easy for readers to follow the analysis. The use of themes, direct quotes, and clear language contributes to the effective communication of the findings. With minor enhancements, such as further elaboration on conflicting views and brief summaries of key points, the results section would be even more compelling.
Research Conclusion section
1. The conclusions of this study are well-supported by the results. They accurately reflect the key findings and are consistent with the data collected through stakeholder engagement. With some enhancements, particularly in providing more specific recommendations and addressing all stakeholder concerns, the conclusions could be made even stronger and more actionable.

Author Response
Dear Reviewer 1,
Thank you very much for taking the time to review our manuscript. Please see the attachment which provides our point-by-point to your comments. In response to your additional comments in the PDF attached to your review report, we have provided our responses below.
Page 2
Comment 1: Make a reference here.
Response 1: We have added in a reference for this sentence, see page 2, line 68 in the revised tracked manuscript.
Comment 2: Could you summarize within one reference.
Response 2: We have amended to summarise these two sentences within the one same reference. See page 2, lines 68-74 of the tracked manuscript.
Comment 3: 1. The author should give more information on why the stakeholder perspective is important for T2D. 2. Do you have the research about No. 1 that is relevant to this topic?
Response 3: We have amended the previous paragraph to strengthen the rationale for focusing on stakeholder perspectives - see page 3, lines 86-105 in the revised tracked manuscript. We also amended section 2.4 to explain the relevance of research question 1 to the study. See page 4, lines 185-187.
Comment 4: What is this study's methodology?
Response 4: We have revised the text to clarify the study's methodology. See page 3, line 119 of the tracked, revised manuscript.
Page 3
Comment 1: Provide the number of participants.
Response 1: We have revised section 3.1 to clarify the full breakdown of the number of participants. See page 5, lines 230-240 in the revised, tracked manuscript
Page 4
Comment 1: Give full abbreviation of KKL.
Response 1: In reporting qualitative studies, it is common practice for the author initials to be provided in the data analysis section without full abbreviation. We have added a sentence to the start of section 2.5 for clarification. See page 5, line 211 in the tracked manuscript revision.
Comment 2: Give full abbreviation of KT, JHYW .
Response 2: In reporting qualitative studies, it is common practice for the author initials to be provided in the data analysis section without full abbreviation. We have added a sentence to the start of section 2.5 for clarification. See page 5, line 211 in the tracked manuscript revision.
Comment 3: Could you please provide the exact number of participants?
Response 3: Section 3.1 has been revised for clarity on the number of study participants, see page 5, lines 230-240 in the revised tracked manuscript. We apologise for the confusion.
Comment 4: Where do the other three come from?
Response 4: In addition to our response to comment 3 above, section 3.1 has also be revised to clarify the sectors represented and where the stakeholders come from. See page 5, lines 230-240 in the revised tracked manuscript. We apologise for the confusion.
Page 11
Comment 1: This methodology should be mentioned early in the Material and Methods section.
Response 1: Thank you, we have amended the material and methods section accordingly, as per our response to your comment 4 on page 2.

Reviewer 2 Report
Comments and Suggestions for Authors
1.
The result section outcome is not clear
The 3 questions in objectibe has been broaden in Table 2-- but while discussing the broder parts the main questions is not discussed
Advised to highlight the major 3 questions after each partwise discussions
may be another combined table need
2.
Findings can be used to inform future research on how Food is Medicine strategies can become a nexus ....- Not cear
3.
The discussion about different food habit relation is not clear.
Author Response
Dear Reviewer 2,
Thank you for taking the time to review our manuscript. Please see the attachment for our detailed point-by-point response to your comments.

Reviewer 3 Report
Comments and Suggestions for Authors
First of all the reviewer congratulates all authors for conducting this context specific study providing evidence for the integration of produce prescription into the health system, given the differences between countries’ health systems.This manuscript is based on a study that explored stakeholders’ acceptability of and perspectives on the potential design, implementation, and integration of produce prescription for adults with type 2 diabetes in Australia. The study was based on key stakeholders identified through purposive sampling based on expertise. Data for the study were collected through a virtual workshop and discussion platform, with workshop questions developed and adapted from prior literature.
According to me the study has been conceived, executed, and presentenced in a scholar manner. The sections on introduction including justification of the research, materials and methods, results and discussions are presented adequately. The conclusion drawn is based on data analysis and discussion.
One of the limitations of the study is that data was collected from the key stakeholders through online Worksop which did not allow for as in-depth discussion among the participants. However, the authors have clearly pointed out these issues in limitation section. The study findings would certainly be contributing to the available literature on the field of study.
In view of the above, I recommend that the manuscript can be considered for publication. However, minor English edition may be required (Example Introduction section line 52 – sentence needs to be corrected).
Comments on the Quality of English Language
Minor editing required
Author Response
Dear Reviewer 3
Thank you very much for taking the time to review this manuscript and recommendation that our submission be considered for publication. Please see the attachment for a point-by-point response to your comments.

Reviewer 4 Report
Comments and Suggestions for Authors
The paper submitted for review is the result of a multi-phase partnership research project to jointly design a sustainable model for implementation partners and a framework for prescription food products for people with T2D. In the reviewer's opinion, the project's design is an absolutely important and crucial scientific activity aimed at the future development of nutritional intervention strategies for people with T2D. From this perspective, the submitted article for review is an important source of information on stakeholder perspectives on the design and implementation of prescription interventions for people with T2D in Australia. Despite my full support for the concept and design of the study, including with regard to the data included in the peer-reviewed article, I have several comments that the authors of the article should critically review and possibly improve or add to:
1/ The title too complicated and difficult to identify the actual concept of the study. The title should be identical with the purpose of the work, which in my opinion is understandable. I propose a title for the article that is identical to the main purpose of the paper ie: Stakeholder perspectives on the design and implementation of prescribing interventions for people with type 2 diabetes in Australia. Stakeholder perspectives include the concepts of acceptability and prescribing concepts and therefore answers to the key research questions posed in the paper.
2/ Please unify the purpose of the work in the abstract of the work so that it is the same as the main purpose in the body of the work.
3/ Due to the very small study sample, 2 to 5 people in each stakeholder group, which may determine a large error in the measurement of perspectives/views due to subjectivity, for example, I propose that the modified title of the paper add: “pilot study” (Stakeholder perspectives on the design and implementation of prescribing interventions for people with type 2 diabetes in Australia - a pilot study). I think this needs to be stated very strongly in the limiting factors, paying particular attention to the subjectivity of opinion, which can lead to erroneous results in assessing stakeholder perspectives. Of particular concern is the number of food retailers, consumers and NGO representatives. These groups may have relatively low knowledge of the problem taken up, as well as being particularly characterized by subjectivity in assessing the situation in the context to the general supopulation of these groups. While the authors of the article argued strongly for limiting factors, including those related to sample size, I believe that stronger arguments from the literature should be used that the sample size is appropriate.
4/ Line 187-194. 21 people participated in the TAG survey (line 188). Finally, 22 people participated in the survey (line191). I don't understand where the difference comes from, please explain!
The absolute strengths of the article are the research method adopted (apart from the sample size, which is somewhat questionable) and the results of the work: results that significantly enrich knowledge of the prospects associated with food prescribing in the context of T2D treatment, and, importantly, enrich knowledge of the potential problems that may arise in the future from such activities. In addition, the results were developed in a broad context that minimizes uncertainties - broadly answering the research questions posed.
The results of the research project, the preliminary results of which are presented in the peer-reviewed article, can be an important source of inspiration for research of this type in other countries, since in my opinion “prescription food” seems to be an inevitable prospect for treatment in the future. Thus, the research topic is very important and requires further discussion.
Author Response
Dear Reviewer 4
Thank you very much for taking the time to review this manuscript. Please see the attachment for our detailed point-by-point responses.
